# Repeated Exposure of Macrophages to Synthetic Amorphous Silica Induces Adaptive Proteome Changes and a Moderate Cell Activation

**DOI:** 10.3390/nano12091424

**Published:** 2022-04-22

**Authors:** Anaelle Torres, Véronique Collin-Faure, Hélène Diemer, Christine Moriscot, Daphna Fenel, Benoît Gallet, Sarah Cianférani, Jacques-Aurélien Sergent, Thierry Rabilloud

**Affiliations:** 1Chemistry and Biology of Metals Laboratory, Université Grenoble Alpes, Centre National de la Recherche Scientifique, Commissariat à l’Energie Atomique, Interdisciplinary Research Institute of Grenoble, 38054 Grenoble, France; anaelle.torres@cea.fr (A.T.); veronique.colllin@cea.fr (V.C.-F.); 2Laboratoire de Spectrométrie de Masse BioOrganique (LSMBO), Centre National de la Rech erche Scientifique, Hubert Curien Pluridisciplinary Institute UMR 7178, Strasbourg University, 67087 Strasbourg, France; hdiemer@unistra.fr (H.D.); sarah.cianferani@unistra.fr (S.C.); 3Infrastructure Nationale de Protéomique ProFI—FR2048, 67087 Strasbourg, France; 4Integrated Structural Biology Grenoble (ISBG), European Molecular Biology Laboratory Université Grenoble Alpes, Centre National de la Recherche Scientifique, Commissariat à l’Energie Atomique, 71 Avenue des Martyrs, 38042 Grenoble, France; christine.moriscot@ibs.fr; 5Institute of Structural Biology (IBS), Université Grenoble Alpes, Centre National de la Recherche Scientifique, Commissariat à l’Energie Atomique, Interdisciplinary Research Institute of Grenoble, 38044 Grenoble, France; daphna.fenel@ibs.fr (D.F.); benoit.gallet@ibs.fr (B.G.); 6Toxicological and Environmental Risk Assessment Unit, Solvay SA, 1120 Brussels, Belgium; jacques-aurelien.sergent@solvay.com

**Keywords:** synthetic amorphous silica, macrophages, proteomics, inflammation, repeated exposure

## Abstract

Synthetic amorphous silica (SAS) is a nanomaterial used in a wide variety of applications, including the use as a food additive. Two types of SAS are commonly employed as a powder additive, precipitated silica and fumed silica. Numerous studies have investigated the effects of synthetic amorphous silica on mammalian cells. However, most of them have used an exposure scheme based on a single dose of SAS. In this study, we have used instead a repeated 10-day exposure scheme in an effort to better simulate the occupational exposure encountered in daily life by consumers and workers. As a biological model, we have used the murine macrophage cell line J774A.1, as macrophages are very important innate immune cells in the response to particulate materials. In order to obtain a better appraisal of the macrophage responses to this repeated exposure to SAS, we have used proteomics as a wide-scale approach. Furthermore, some of the biological pathways detected as modulated by the exposure to SAS by the proteomic experiments have been validated through targeted experiments. Overall, proteomics showed that precipitated SAS induced a more important macrophage response than fumed SAS at equal dose. Nevertheless, validation experiments showed that most of the responses detected by proteomics are indeed adaptive, as the cellular homeostasis appeared to be maintained at the end of the exposure. For example, the intracellular glutathione levels or the mitochondrial transmembrane potential at the end of the 10 days exposure were similar for SAS-exposed cells and for unexposed cells. Similarly, no gross lysosomal damage was observed after repeated exposure to SAS. Nevertheless, important functions of macrophages such as phagocytosis, TNFα, and interleukin-6 secretion were up-modulated after exposure, as was the expression of important membrane proteins such as the scavenger receptors, MHC-II, or the MAC-1 receptor. These results suggest that repeated exposure to low doses of SAS slightly modulates the immune functions of macrophages, which may alter the homeostasis of the immune system.

## 1. Introduction

Synthetic amorphous silica (SAS) is one of the most produced (and used) nanomaterials, with a worldwide production in millions of tons per year [1]. Due to this high production and use, the potential adverse effects of this nanomaterial need to be investigated in detail. This investigation is made even more acutely needed by the fact that another form of silica, namely crystalline silica, is known to be the etiological agent of a severe disease, silicosis [2,3]. Silicosis is generally linked to working conditions such as mining or sandblasting, but may also occur through environmental exposure [4]. Silicosis being a chronic inflammatory disease, innate immune cells such as macrophages are at the root of the disease’s etiology [5,6,7,8,9,10]. Based on this paradigm, studies of the toxicological impact of SAS have focused on the persistence of the effects on in vivo model [11,12,13,14] and on macrophage models for in vitro systems (as reviewed for example in [15,16]). These studies have shown that SAS is selectively toxic for macrophages [17,18] but not for other cell types [19], and also induces a pro-inflammatory response [20,21,22,23]. This suggests a strong parallel between the toxicology of SAS and the one of crystalline silica. However, the in vitro studies that have investigated the pro-inflammatory effects of SAS have used a classical acute exposure mode, and thus cannot investigate the persistence of the effects, which is a key dimension in the etiology of silicosis. In vivo studies have shown that the effects of SAS are transient [11,12,13,14], which has been recently transposed (and confirmed) using dedicated in vitro systems [24].

When assessing the toxicology of SAS, another point to be taken into account is that the exposures to this nanomaterial are generally chronic, i.e., repeated and at low concentrations over a long period, e.g., through the use of SAS as a food additive or in cosmetics. This is a critical dimension to consider when studying the effects of SAS on macrophages in vitro. Indeed, the few studies that have investigated the effects of repeated exposures, mostly on silver nanoparticles, have reported different effects than those observed after an exposure to a single dose [25,26,27,28,29,30]. Such studies are much less common for silica [31] and have centered more on the structural determinants of the effects of silica on cells rather than on the cellular responses. This is an active area of research [32,33,34,35], as the chemical determinants of the effects of silica on cells are much less obvious than for metallic nanoparticles, which often produce their effects through partial dissolution and liberation of toxic metal ions [36].

In complement to the studies devoted to the understanding of the chemical features that drive silica toxicity, we decided to carry out a study centered on the cellular responses of macrophages to SAS. We thus chose to investigate the effects of two different types of SAS, namely precipitated silica and fumed silica. Precipitated silica is prepared via a wet route starting from alkaline silicates in water, while fumed silica is prepared via a pyrolytic route starting from silicon halides that are burnt in a flame in the presence of oxygen. Both types of silica are produced and sold as solids and can be used in a variety of applications ranging from tires and resins reinforcement to additives in food and cosmetics. As exposures to these forms of SAS are typically occupational, we chose to investigate the effects of a repeated exposure on macrophages, using a combination of proteomics and targeted studies. Omics studies in general and proteomics in particular have been shown to be interesting tools to better decipher the cellular responses to nanomaterials [28,29,37,38], sometimes allowing us to unravel new mechanisms at play in the effects of nanomaterials [28,37,38].

## 2. Materials and Methods

### 2.1. Nanomaterials and Reagents

The precipitated SAS used in this study was provided based on a collaboration with Solvay (Solvay GBU Silica, Lyon, France), reflecting a commercial grade of SAS. The primary particle size was in the 10–20 nm range, the agglomerate size in the micron range, and the specific surface area close to 220 m^2^/g. The fumed silica used in this study was purchased from Sigma (catalog #S5055, St. Louis, MI, USA). Its primary particle size was in the 20–30 nm range, the agglomerate size in the 250-nm range, and the specific surface area close to 200 m^2^/g. More characterization details have been provided in previous publications [24,39] and are also available in Appendix A Appendix A. For use in cell cultures, both forms of silica were suspended in ultrapure water at the concentration of 10 mg/mL and pasteurized overnight after 10 min of sonication in an ultrasonic bath.

Unless otherwise stated, all reagents were at least 99% pure and purchased from Sigma (St. Louis, MO, USA)

### 2.2. Cell Culture

The mouse macrophage cell line J774A.1 was obtained from the European Cell Culture Collection (Salisbury, UK). The cells were cultured in DMEM medium +10% fetal bovine serum (FBS) under an air 5% CO_2_ atmosphere. For routine culture, cells were seeded on non-adherent flasks (e.g., suspension culture flasks from Greiner) at 200,000 cells/mL and harvested 48 h later, at 1,000,000 cells/mL. Cell viability was measured by a dye exclusion assay, either with eosin (1 mg/mL) under the microscope or with propidium iodide (1 µg/mL) in a flow cytometry mode.

For treatment with SAS, cells were seeded at 500,000 cells/mL and cultured for 24 h at 37 °C for cell adhesion. The medium volume was adjusted to keep a constant medium height for all the culture supports used (2 mL for six-well plates and 15 mL for 75 cm^2^ flasks). The cells were then grown to confluence for 48 h. The cells were then treated daily with 2 µg/mL silica for ten days, with a culture medium change every two days. In order to take cell detachment into account [40], the medium removed at each medium changed was centrifuged (100 g, 5 min) to collect the detached cells. The cell pellets obtained at this stage were resuspended in the fresh culture medium and added to the remaining cell layer. At the end of the exposure period, the cell culture medium was removed and the cells were collected by scraping in PBS, then rinsed twice in PBS and processed. In order to investigate cell responses to bacteria, stimulation with LPS (100 ng/mL) was carried out in some experiments for the last 24 h in culture before harvesting. When an exposure to fluorescent colloidal silica (Micromod Sicastar green ref 42.00.301) was carried out, the cells were exposed for the last 24 h in culture to a concentration of 20 µg/mL of fluorescent silica.

### 2.3. SAS Quantification by ICP-AES

#### 2.3.1. Sample Preparation

For SAS internalization and quantification in the cells, the cells were grown on adherent six-well plates and exposed to silica as described above. One milliliter of lysis solution (5 mM HEPES pH 7.5, 0.75 mM spermine tetrahydrochloride, and 0.1% tetradecyldimethylammonio propane sulfonate) was added in the wells, after the medium had been removed and the cell layer washed once with PBS. The lysates were aliquoted and frozen. The day before the quantification by ICP-AES, the samples were mineralized: in each sample, an equal volume of 1 N potassium hydroxide was added; the samples were then incubated overnight at 80 °C in a preheated water bath [41].

#### 2.3.2. ICP-AES Dosage

The multi-elemental standards for ICP were purchased from Sigma-Aldrich (ref 92091) and were first prepared in 10% HNO_3_. At the end of the standard measurement, samples were prepared two at a time (base-mineralized silica is not stable in HNO_3_). From 250 to 500 µL of mineralisate was added to 10% HNO_3_ to reach a final volume of 6 mL. The samples were measured in an ICP-AES Shimadzu 9000 with ICPE solution Launcher software (v1.31, Shimadzu France, Noisiel, France).

### 2.4. Proteomics

#### 2.4.1. Sample Preparation

The cell pellets obtained after exposure to SAS (or from mock-exposed cells cultivated under the same conditions) were lysed in were extracted with 10 pellet volumes of lysis solution (7 M urea, 2 M thiourea, 15 mM spermine base, 15 mM spermine tetrahydrochloride, 10 mM Tris (carboxyethyl) phosphine hydrochloride and 4% (*w*/*v*) CHAPS). After extraction at room temperature for one hour, the extracts were clarified by centrifugation (15,000× *g* 15 min), the supernatants collected, and their protein concentration determined by a modified Bradford assay [42]. The protein extracts were stored frozen at −20 °C until use.

#### 2.4.2. Sample Processing and Mass Spectrometry

For the shotgun proteomic analysis, the samples were included in polyacrylamide plugs according to Muller et al. [43], with some modifications to downscale the process [44]. To this purpose, the photopolymerization system using methylene blue, toluene sulfonate, and diphenyliodonium chloride was used [45].

First, a sample dilution solution (8 M urea in 200 mM Tris-HCl, pH 8) was prepared. Then, the dye solution (0.5 mM methylene blue in sample dilution solution) was prepared. The other initiator solutions consisted of a 1 M solution of sodium toluene sulfinate in water and in a saturated water solution of diphenyliodonium chloride. The ready-to-use polyacrylamide solution consisted of 1.2 mL of a commercial 40% acrylamide/bis solution (37.5/1) to which 100 µL of diphenyliodonium chloride solution, 100 µL of sodium toluene sulfinate solution, and 100 µL of water were added.

Protein samples (20 µg each) were first diluted to a 9 µL final volume with sample dilution solution in a 500 µL conical microtube (Eppendorf, Hamburg, Germany). To this solution, 6 µL of dye solution then 5 µL of acrylamide solution were then added and mixed by pipetting. Then, 100 µL of water-saturated butanol were layered on top of the samples, and polymerization was carried out under a 1500 lumen 2700 K LED lamp for 2 h, during which the initially blue gel solution discolored. At the end of the polymerization period, the butanol was removed, and the gel plugs were fixed for 2 × 1 h with 200 µL of 30% ethanol 2% phosphoric acid, followed by a 30-min wash in 30% ethanol. The fixed gel plugs were then stored at −20 °C until use.

Gel plug processing, digestion, peptide extraction, and nanoLC-MS/MS was performed as previously described [44], without the robotic protein handling system and using a Q-Exactive Plus mass spectrometer (Thermo Fisher Scientific, Bremen, Germany).

#### 2.4.3. Protein Identification

For protein identification, the MS/MS data were interpreted using a local Mascot server with MASCOT 2.6.2 algorithm (Matrix Science, London, UK) against an in-house database containing all Mus musculus and Rattus norvegicus entries from UniProtKB/SwissProt (version 2019_10, 50,313 sequences) and the corresponding 50,313 reverse entries. Spectra were searched with a mass tolerance of 10 ppm for MS and 0.07 Da for MS/MS data, allowing a maximum of one trypsin missed cleavage. Trypsin was specified as an enzyme. Acetylation of protein n-termini, carbamidomethylation of cysteine residues, and oxidation of methionine residues were specified as variable modifications. Identification results were imported into Proline software version 2.1 (http://proline.profiproteomics.fr/) for validation. Peptide Spectrum Matches (PSM) with pretty rank equal to one were retained. False Discovery Rate was then optimized to be below 1% at PSM level using Mascot Adjusted E-value and below 1% at Protein Level using Mascot Mudpit score.

Mass spectrometry data are available via ProteomeXchange [46] with the identifier PXD03002, associated with the doi: 10.6019/PXD030002.

#### 2.4.4. Label Free Quantification

Peptide Abundances were extracted using the Proline software version 2.1 (http://proline.profiproteomics.fr/) with a *m*/*z* tolerance set at 10 ppm. Alignment of the LC-MS runs was performed using Loess smoothing. Cross Assignment was performed within groups only. Protein Abundances were computed by sum of peptide abundances (normalized using the median).

#### 2.4.5. Data Analysis

For the global analysis of the protein abundances data, we first excluded all proteins identified by only one peptide. The protein abundances data were normalized to the sum of all protein abundances, converted to ppb (part per billion), and missing data were imputated with a low, non-null value. The relative abundances data were used directly for global analysis using the PAST software suite [47] without any transformation. Principal correspondence analysis and non-metric multidimensional scaling were used to assess the global differences between samples. In order to minimize the quantitative bias due to the widely different protein abundances within each sample, the Gower distance (i.e., a normalized distance) was used to perform the principal correspondence analysis.

Proteins were considered as significantly different if their U value in the Mann–Whitney U test was ≤2 in the control vs. SAS-treated comparison. As the proteomics experiments were performed on five independent biological replicates, this corresponded to *p* < 0.05. No quantitative change threshold value was applied. However, a false discovery rate was calculated for each selected protein, using the sequential Fisher test approach [48]. The selected proteins were then submitted to pathway analysis using the DAVID tool [49], with a FDR cutoff value set at 0.1.

### 2.5. Phagocytosis and Particle Internalization Assay

The phagocytic activity was measured using fluorescent latex beads (1 µm diameter, green labelled, catalog number L4655 from Sigma), with exclusion of the dead cells from the analysis, as described previously [50].

### 2.6. Mitochondrial Transmembrane Potential Measurement

The mitochondrial transmembrane potential was assessed by Rhodamine 123 uptake at low concentration (80 nM) to avoid quenching [51], as described previously [52].

### 2.7. Reduced Glutathione Measurement

The reduced glutathione was assessed by using monochlorobimane at 70 µM final concentration, with exclusion of the dead cells from the analysis, as described previously [53]. This experiment was carried out with a FacsMelody flow cytometer and the FacsChorus software (v1.3.3. BD Biosciences, Le Pont de Claix, France) for data analysis.

### 2.8. Lysosomal Function Evaluation

The ratiometric acridine orange fluorescence was used to investigate the lysosomal function. Cells were seeded into six-well plates and exposed to silica, as described above. Acridine orange (Sigma A6529) was added to the cell culture (100 ng/mL), and the culture returned to the incubator for 30 min. Then, cells were harvested and washed with PBS-glucose (PBSG). The pellets were suspended with 250 μL of PBSG supplemented with Sytox Red (5 nM), and analyzed by flow cytometry. Acridine orange was excited at 475 nm, and the 526 nm and 650 nm emissions were recorded. The ratio between the two fluorescence intensities (Red/Green) was then calculated and used as an index of the lysosomal function [54].

### 2.9. Surface Markers

The cell surface labelling was used as a marker of cellular status. Cells were seeded into six-well plates and exposed to SAS for 10 days, as described above. Cells were harvested, washed with PBS, and fixed for 30 min with paraformaldehyde 4% at room temperature. Cells were washed again and stored at 4 °C until the cell surface labelling. The fixed cells were collected and dispersed in a 96-well plate with 300,000 cells per well. After a step of washing and centrifugation (1 min at 800× *g*), the cells were incubated 10 min at room temperature with PBS-fluo (PBS 1X containing 0.16% sodium azide and 3% of FBS) and Fc-block (purchased from eBiosciences, dilution 1/50). The cells were centrifuged and then incubated 20 min protected from light with PBS-fluo containing antibody of interest (CD14-APC: BD ref 560634 dilution 1/200, CD38-FITC: BD ref 558813 dilution 1/200, CD86-APC: BD ref 558703 dilution 1/200, MARCO-FITC: R&Dsystems ref FAB2956F dilution 1/25, CD11b-FITC: eBiosciences ref dilution 1/100, CD204-PE: Invitrogen ref 12-2046-82 dilution 1/100, CD18-APC: BD Pharmingen ref 562828 dilution 1/50). Then, cells were washed twice with PBS-fluo, and finally, the cell pellets were suspended in 200 µL of PBS fluo for FACS analysis.

(FITC: excitation 488 nm emission 525 nm, APC: excitation 633 nm emission 760 nm, PE: excitation 488 nm, emission 575 nm.)

### 2.10. Cytokines and Nitric Oxide Production

Cells were grown into six-well plates and exposed to the indicated dose of SAS for 10 days at 37 °C. The cells were also primed or not with LPS (100 ng/mL) for the last 24 h of culture. For nitric oxide production, 5 mM arginine monohydrochloride were added for the last 24 h of culture, in order to provide unlimiting substrate for the NO synthase. The supernatants are collected and centrifuged to eliminate non-adherent cells. For nitric oxide, the nitrite concentration of the supernatants was measured with the Griess reagent at 540 nm. For cytokines, the experiments were carried out with the Cytometric Bead Array Mouse Inflammation Kit (BD Biosciences) and analyzed with FCAP Array software (3.0, BD Biosciences, Le Pont de Claix, France). This Flex-set kit allows measuring Interleukin-6 (IL-6) and TNFα protein levels in a single sample. The mixed capture beads were added to all assay tubes containing supernatant samples and standards (from 0 to 2500 pg/mL), the mouse inflammation phycoerythrin (PE, excitation 488 nm, emission 575 nm) detection reagent was added, and the mixture was incubated for 2 h at room temperature, protected from light. The wash buffer was added to each tube, which were then centrifuged 5 min at 200× *g*, and the pellets were resuspended with the wash buffer and analyzed by FacsCalibur flow cytometer.

## 3. Results

### 3.1. Toxic Effects of Repeated Exposure to Synthetic Amorphous Silica

First, we checked that the repeated treatment with either form of SAS was not toxic to the cells. Indeed, the dose was determined from a previous work with fumed silica [24], in which an acute LD20 dose was determined to be 20 µg/mL for 24 h. Thus, we fractionated the dose in 10 subdoses of 2 µg/mL each. As shown in Figure 1, this treatment had no visible toxicity on macrophages.

### 3.2. Silica Uptake

Silica uptake was measured after the 10 days exposure period and also after a 24 h exposure to the same cumulative dose (1 × 20 µg/mL vs. 10 × 2 µg/mL/day). The results, shown in Table 1, indicate a significant uptake of silica in exposed macrophages. Interestingly, the uptake was more pronounced for precipitated silica than for fumed silica in the case of the 10 days exposure. This may be linked to the different aggregate sizes observed for the two forms of SAS after being in contact with proteins, as is the case for complete culture medium (Appendix A).

Moreover, the uptake was much lower in the case of the 10 days exposure than in the 24 h exposure. This phenomenon has also been described for silver nanoparticles [29], but not in the case of a short-term (4 days) repeated exposure to colloidal silica [50].

Regarding silica uptake, we also checked by confocal and electron microscopy that, even after 24 h of exposure, silica was present mostly inside the cells and not only at their surface, which could have biased the uptake measurements (Appendix A).

### 3.3. Global Analysis of the Proteomic Results

When the raw proteomic data were filtered for proteins identified and quantified by at least two independent peptides, 2322 proteins were selected. As a first step in the exploitation of the proteomic data, a global analysis was performed. The purpose of such an analysis is to use all the protein abundance data (Appendix A) to determine the intensity of the global effects of the repeated exposures to precipitated and fumed silica.

The results, shown in Figure 2, indicated that precipitated silica had an overall greater effect on macrophages than fumed silica. The first two axes accounted for 35 and 10% of the variance, respectively.

In order to take all the information into account, non-metric multidimensional scaling was used. The results, shown in Appendix A, confirmed the results of the principal correspondence analysis, and thus that precipitate silica induced more proteome changes than fumed silica.

In a second step, the protein showing significant expression changes (*p* < 0.05, i.e., U ≤ 2 in the Mann–Whitney U test) between the exposed and control cells was selected, resulting in two separate lists for the proteins modulated by the treatment with precipitated silica (Appendix A) and the proteins modulated by the treatment with fumed silica (Appendix A). Regarding numbers, 287 proteins were modulated in response to fumed silica, while 814 proteins were modulated in response to precipitated silica. Of all these proteins, 209 were modulated by both treatments, leaving 78 proteins modulated only in response to fumed silica and 605 modulated only in response to precipitated silica.

Both lists were used to perform pathways analyses by the DAVID software (update 2021, Frederick National Laboratory for Cancer Research, Frederick, MD USA). The results of the pathway analyses are detailed in Appendix A, respectively. Some highlighted pathways indicated a global stress response (e.g., carbon metabolism, splicing, translation), which is expected for any cellular stress, while other pathways (e.g., lysosome, mitochondrion, phagosome, antigen presentation, immunity) were more specific. The main pathways highlighted by the DAVID software are summarized in Figure 3.

Precipitated synthetic amorphous silica is responsible for more protein changes in cell metabolism, function and binding than fumed SAS. Both these specific pathways and the proteins highlighted in them by the software were selected for detailed analysis.

### 3.4. Detailed Analysis of the Proteomic Results and Validation Experiments

In addition to the results of the pathway analysis, a manual analysis of the modulated protein lists was performed. This allowed for the selection of interesting proteins that escaped the pathway analysis because of either their poor annotation or the fact that they did not belong in an enriched-enough pathway to be selected by the software. Based on these data, we focused our detailed analyses on a few subsets of proteins

#### 3.4.1. Mitochondrial Proteins

Seventy-six mitochondrial proteins were found to be modulated by the pathway analysis software (Appendix A). Of these, 32 were decreased upon treatment with precipitated silica while 44 were increased, with a median fold change (in both directions) of ±21%. Furthermore, 14 of these proteins were also modulated in response to exposure to fumed silica. These numbers indicated rather minor, adaptive changes. For example, the 27 kDa subunit of the MICOS complex (accession number Q78IK4), implied in mitochondrial architecture, was decreased by 10% upon treatment with both forms of SAS. However, the 13, 19, 25, and 60 kDa subunits were also detected in the proteomic screen and did not show any significant change in response to either treatment. This further suggested small amplitude and adaptive changes. To test this hypothesis, we measured the mitochondrial transmembrane potential. The results, shown in Figure 4A,B, indicated an absence of alteration of the transmembrane potential, supporting the hypothesis of adaptive changes in the mitochondrial proteins.

#### 3.4.2. Glutathione Levels

“Cell redox homeostasis” was among the functional clusters highlighted by the pathway analysis, with 17 proteins modulated (Appendix A). Of these, five were also modulated in response to exposure to fumed silica, and one (glutathione reductase) was modulated in response to exposure to fumed silica, but not precipitated silica.

The modulated proteins included five of the six peroxiredoxins, i.e., proteins involved in the destruction of peroxides, but also two proteins involved in glutathione metabolism, namely glutathione synthase and glutathione reductase. To check whether the repeated exposure to synthetic amorphous silica may induce perturbations in the redox balance, we measured the level of intracellular reduced glutathione. The results, shown in Figure 4C,D, indicated no change in the intracellular glutathione level.

#### 3.4.3. Lysosomes and Phagocytosis

Forty-four lysosome-associated proteins were found modulated by the treatment with precipitated silica (Appendix A). Of these, 12 were also modulated in response to exposure to fumed silica. Interestingly, the luminal lysosomal proteins (e.g., cathepsins B, D, S and Z, alpha and beta mannosidases, arylsulfatases A and B) showed decreased abundances upon treatment, with a moderate median change of −30%, while the few lysosomal membranes detected (Q9CQW9, P52875, Q9Z0M5, P24668) were increased. This prompted us to test the lysosomal integrity by the acridine orange ratiometric method [54]. The results, shown in Figure 5A, indicated no change in the 650 nm/526 nm fluorescence, thereby indicating no gross alteration of the lysosomes upon treatment with either form of SAS.

In macrophages, lysosomes are also associated with the phagocytic function, and this link was highlighted in the pathway analysis by functional clusters such as “actin cytoskeleton”, but also by the modulation of subunits of the proton ATPase (P50408, P50518, P51863, P63081, Q9CR51), of granulins (P28798), and of activators of the NADPH oxidase (O70145, Q09014). We therefore probed the phagocytic function by flow cytometry. The results, shown in Figure 5B,C, indicated that the proportion of phagocytic cells in the culture remained constant, even after the treatment with either form of synthetic amorphous silica. However, detailed analysis of the fluorescence pattern showed a slight but significant increase in the phagocytic capacity of silica-treated cells over untreated cells.

#### 3.4.4. Immunity-Associated Proteins

Thirty proteins associated with the keyword “immunity” were found to be modulated by the treatment with precipitated silica. (Appendix A). In addition to these, other immunity-associated proteins, and especially surface markers such as CD11b, CD18, CD86, or CD204, were also found significantly modified in their abundances upon treatment with precipitated silica, as well as fumed silica for CD11b and CD18. While CD11b and CD 18 showed a moderate decrease in their abundances upon treatment with precipitated silica (−11% and −15%, respectively), CD14 showed a moderate increase in abundance (+11%), while CD 86 and CD204 showed a higher increase in their abundances (+20% and +40%, respectively). In order to validate these proteomics results, a flow cytometry analysis of the surface expression of these markers was carried out. In addition to the markers detected by proteomics (CD11b, CD14, CD18, CD74 = MHC Class II, CD86, and CD204), we also tested a marker of M1 polarization (CD38) [55,56] and another form of scavenger receptor (MARCO) [57,58], and the results are shown in Figure 6. They indicated a slight but significant increase in the surface expression of CD11b and CD18 and an important increase in the proportion of CD14-positive cells after treatment with both forms of synthetic amorphous silica. An important increase in the proportion of positive cells was also observed in response to exposure to both forms of synthetic amorphous silica for CD38, MARCO, and CD74 (MHC class II). For CD86, the proportion of positive cells was slightly but significantly decreased in response to fumed silica but not to precipitated silica. However, the signal of the positive cells was significantly lower in cells exposed to both forms of silica than in control cells.

Regarding CD204, a slight but significant increase in its surface expression was observed after exposure to precipitated silica but not to fumed silica.

As scavenger receptors are known to be the receptors of silica [59,60,61], we checked whether this slight overexpression could be linked to a saturation of the silica internalization process after 10 days of continuous exposure. To this purpose, we treated macrophages for nine days with the SAS of interest, and exposed them to fluorescent colloidal amorphous silica for the last day. The results, shown in Figure 7, indicated that silica-treated cells internalized more fluorescent colloidal silica than untreated cells, which correlated with the increased expression of CD204 and MARCO.

As proteomics cannot easily test secreted proteins because of their high efflux and low steady state levels in cells, we decided to test some pro-inflammatory cytokines. We thus tested the secretion of NO, TNFα, and IL-6 in response to the repeated exposure to synthetic amorphous silica, with and without terminal LPS stimulation. The results, shown in Figure 8, indicated a slight but significant increase in NO and TNFα production after treatment with SAS. In the case of TNFα, a more pronounced increase was observed after treatment with fumed silica compared to precipitated silica. This increase was also observed after LPS stimulation and was just transposed in range from the high pg/mL range (without LPS) to the medium ng/mL range with LPS. Regarding IL-6, its secretion remained undetectable (i.e., below 5 pg/mL) without LPS stimulation. With LPS stimulation, an increase in LPS secretion was observed after repeated exposure to fumed silica, but not to precipitated silica.

## 4. Discussion

Synthetic amorphous silica is used in a wide variety of applications and comes in three main forms, i.e., colloidal silica, precipitated silica, and fumed silica. Opposed to colloidal silica, which is by definition a suspension of silica nanoparticles in water, precipitated and fumed silica are produced as powders. Consequently, exposure to their dusts may occur in workers and in the general public, depending on their modalities of use. Such exposures are, by definition, repeated exposures, and it was thus necessary to use a repeated exposure scheme to reproduce the in vivo reality as closely as possible in our in vitro experiments. Indeed, it has been shown in the literature that repeated exposures produce a different biological outcome than single dose exposures, as described on silver nanoparticles [25,28,29] and on colloidal silica [50].

The dose that we have applied (2 µg/mL daily) has also been used in the literature [31] and considered as a realistic dose, in line with the typical exposure at the workplace or by consumers. Contrasting with this publication, we focused on the amorphous silica “as present and marketed”, i.e., without forced disaggregation, compared two types of commercial SAS (precipitated and fumed) with similar descriptors, and used proteomics to get a wider description of the macrophages responses to silica.

Although the basic parameters of the two SAS compared in this study were very similar (specific surface area of ca. 200 m^2^/g, primary particle size of ca. 20 nm, aggregate size in the 200–500 nm range), the biological responses, as revealed by proteomics, were significantly different. The precipitated silica induced a much stronger response, as indicated by the number of proteins showing a significant change in their amounts (938 for precipitated silica, 342 for fumed silica). Of those, 238 were significantly changed by both types of silica. The differences in numbers and thus in the extent of the cellular response may be correlated with the differences observed in silica uptake, which was higher for precipitated silica than for fumed silica, a phenomenon that was also found in the case of a single exposure for 24 h. This different uptake may be due either to a different intrinsic uptake due to structural differences between both types of silica or to a slightly different sedimentation behavior, leading to different amounts reaching the cell surface in culture.

Regarding silica uptake, the fact that the uptake was significantly lower in the case of repeated exposure compared to a single exposure to the same cumulative dose was worth investigating. It may be due to a culture artifact linked to cell proliferation and thus dilution via cell division. However, previous data obtained on the J774A.1 cell line suggested that a cell density plateau is reached after four days in culture [40], so that this hypothesis cannot explain the difference between acutely and repeatedly-exposed cells. It may be caused by a lesser internalization capacity induced by the repeated exposure or by an elimination mechanism. As the former hypothesis would go against the increased surface expression of the scavenger receptors CD204 and MARCO, we specifically tested silica uptake at the end of the repeated exposure with a fluorescently-labelled colloidal silica and found an increased uptake compared to control cells, i.e., in line with the surface receptors. This suggests that repeatedly-exposed cells are able to eliminate some of the internalized silica, maybe by dissolution or by vomocytosis [62]. However, we could not detect silicon release in the cell culture medium after exposure to SAS (Appendix A), which may be simply linked to a lack of sensitivity of the technique compared to the low amounts of silicon to be detected.

Pathway analysis of the proteomic-detected changes revealed that very different pathways were altered. Some were ubiquitous, in the sense that they are part of the homeostasis of many different cell types. Pathways such as “endoplasmic reticulum”, “mitochondria”,” cell redox homeostasis”, or “lysosome” belong to this class. As this may indicate a major dysfunction of the cells after this repeated exposure to silica, validation experiments were carried out. They indicated no major cellular dysfunction, which was in line with the low toxicity of amorphous silica upon repeated exposure [31,50]. Thus, the responses observed by proteomics can be described as successful adaptive changes of the proteome, resulting in cell adaptation to the repeated exposure to silica.

In this frame, it was relevant to dissect in greater detail the changes highlighted by the proteomic analysis. A good example is represented by the lysosome, which was highlighted as modified when the cells were treated with precipitated silica. Indeed, most of the luminal lysosomal proteins were decreased upon the treatment with silica by a factor varying from 1.2 to 2 (mean 1.47), while the membrane lysosomal proteins were increased by a factor varying from 1.1 to 1.7 (mean 1.35). One explanation to this phenomenon may be an adsorption of the luminal proteins to the silica present in the lysosomes (where silica is known to be located [63]), thereby decreasing their extractability for the proteomic analyses.

The situation was very different for the mitochondrial proteins. For this protein subset, on the 87 proteins for which the abundance changed upon treatment with precipitated silica, 39 showed a decreased abundance (mean fold change 1.4) and 48 showed an increased abundance (mean fold change 1.25). Interestingly, the ATP synthase and ADP/ATP translocase were increased, as well as mitofusin, i.e., a protein that controls mitochondrial fusion [64]. Conversely, the protein import machinery proteins were decreased, as well as the master regulator Myg1 [65], by factors of 1.2–1.25. As the cell viability and the mitochondrial transmembrane potential were not different from those of the control cells, these changes point to a fine tuning of the proteome to keep the mitochondrial function homeostatic.

The toxicology of amorphous silica has been described as involving oxidative stress [20,66,67]. In this frame, it was logical that the cluster “redox cell homeostasis” appeared in the pathway analysis. A major protein family appearing in this cluster was that of peroxiredoxins and their indirect partner thioredoxin reductase, which is implicated in their activity [68] and is increased in response to precipitated silica. Of the six peroxiredoxins known to date, only Prdx2 was not modulated in response to the treatment with silica. Peroxiredoxins 5 and 6 were found to be reduced. In the case of peroxiredoxin 6, this may be linked with its lysosomal localization [69], as discussed above. For peroxiredoxin 5, which has a complex intracellular localization [70], the situation is less clear. The other three peroxiredoxins, namely Prdx 1, 3, and 4, were increased in response with precipitated silica (Prdx3) or both forms of silica (Prdx 1 and 4). Prdx3 being a mitochondrial protein, its induction, although moderate, suggests an increased oxidative stress at the mitochondrial level. The induction of Prdx4 is worth discussing. In the endoplasmic reticulum, Prdx4 plays a role in protein oxidative folding [71]. Thus, its induction suggests a mechanism by which cells maintain their redox balance in the endoplasmic reticulum, which is critical for the production of membrane and secreted proteins. Moreover, Prdx4 is known not to be fully retained in the reticulum, so that part of it is secreted. Thus, the increase that we observed through proteomics, i.e., the increase in the intracellular fraction, may also indicate an increase in the secreted fraction. As secreted Prdx4 has been implicated in the control of inflammation [72], this may represent one mechanism by which macrophages control their inflammatory response following exposure to silica.

Apart from changes in the peroxiredoxin system, we also detected changes in the amounts of proteins implicated in glutathione metabolism, namely glutathione synthase and glutathione reductase. As decreases in reduced glutathione levels have been reported in macrophages acutely treated with silica [20,67], we investigated the levels of reduced glutathione in cells repeatedly treated with synthetic amorphous silica and found no decrease, indicating that the changes observed in the proteins controlling the cellular redox status can be viewed as successful adaptive changes.

Regarding the surface markers, the first, technical point that needs to be discussed is the discrepancy that can sometimes occur between the proteomic results and the flow cytometry ones. For example, proteomics detected a decrease in the amount of CD11b and CD18 in response to cell exposure to silica, while flow cytometry detected an increase. The converse was true for CD86. In our opinion, this can be simply explained by the fact that both approaches do not measure the same parameter. Proteomics measures the global expression within the whole cell, without distinguishing the various forms of the protein (e.g., precursor vs. mature forms), while flow cytometry in the format used measures the surface expression of the protein. Thus, either intracellular turnover phenomena or masking/unmasking of the protein at the cell surface may explain this discrepant result. This also further demonstrates, if needed, the requirement for validation of proteomic data by functional assays.

This stated, the surface expression data suggested that repeated exposure to synthetic amorphous silica induced an increased surface expression of scavenger receptors, so that the macrophages become prepared to ingest more silica. This is also consistent with the increased phagocytic capacity that was observed after the repeated exposure to silica.

Secondly, the data pointed to a M1 polarization, as exemplified by the increased expression of CD38 [56] and the increased production of NO and pro-inflammatory cytokines (e.g., TNFα), and also suggested some form of macrophage activation. The data are, however, not univocal: CD74 (MHC-II) is increased, as well as the expression of CD11b, which has been reported to be pro-inflammatory [73], although an anti-inflammatory role of CD11b has also been reported [74]. In addition, the co-stimulatory molecule CD86 is decreased. Nevertheless, this response is similar to the silica-driven activation that has been described for dendritic cells [75,76,77], i.e., a closely-related immune cell type.

Regarding the cytokinic inflammatory responses, comparing the responses after an acute exposure or a repeated exposure could be interesting. To this purpose, we used our previous results obtained with the same fumed silica nanomaterial, but on the RAW264.7 model [24] instead of the J774A.1 model used in the present study. While this difference in cell line means that a direct and complete comparison cannot be made, the fact that both cell lines are established murine macrophage models suggests that such a comparison can be relevant.

Within this frame, the increase in TNFα production upon exposure to fumed silica was lower after repeated exposure (1.5-fold) than after acute exposure (2.5-fold). Such a lower increase after a time fractioned dose has also been observed for colloidal silica [50]. The stronger difference was, however, observed for IL-6, which remained undetectable after repeated exposure to both forms of synthetic amorphous silica (this study) while it was clearly detectable (and induced) after an acute exposure to fumed silica [24].

Besides this intrinsic response of the cells to SAS, we also investigated whether the treatment with SAS would modulate the physiological response of macrophages to a bacterial challenge, mimicked by an exposure to LPS. To this purpose, the cells were first exposed repeatedly to SAS and finally to LPS. Our results showed an increased pro-inflammatory response in cells exposed to SAS, suggesting a hyper-reactive state and an exacerbated response. This phenomenon was more pronounced for fumed silica than for precipitated silica.

## 5. Conclusions

In conclusion, proteomics revealed that, although classified both as synthetic amorphous silica, precipitated silica and fumed silica did not induce identical cellular responses after a repeated exposure at non-toxic doses, a phenomenon already described using acute exposure and either precipitated or colloidal silica in comparison with fumed silica [78,79]. Most of the changes detected through proteomics did not reflect functionally as deleterious cellular responses, showing that these changes were adaptive and probably linked to the maintenance of cellular homeostasis. However, macrophage activation and polarization were detected, in line with the well-known pro-inflammatory effects of silica on this cellular type [16,20,22,61,80,81,82], but in acute exposure schemes. It shall be underlined that the pro-inflammatory effects that we detected under these repeated exposure conditions were of lower magnitude than those described after an acute exposure, which further demonstrates, if needed, the importance of the dose rate, even in in vitro toxicology. Indeed, the few repeated exposure experiments that have been carried out in in vitro toxicology [25,27,28,29,31,83] have generally demonstrated different responses when compared to acute exposures. However, the relative effects of both types of exposure cannot be predicted. In the case of silver nanoparticles, short repetition results in smaller effects than acute exposure [26,30], while long repetitions induce stronger effects [28,29,83]. In the case of silica, both short [50] and long (this study), [31] repetitions induce smaller effects than acute exposures. Of note, although the effects detected by proteomics were significantly higher for precipitated silica than for fumed silica, the functional effects, e.g., the effects on surface markers and cytokine production, were similar for both types of silica. This phenomenon may be linked to the common responses detected by proteomics for both types of SAS.

## Figures and Tables

**Figure 1 nanomaterials-12-01424-f001:**
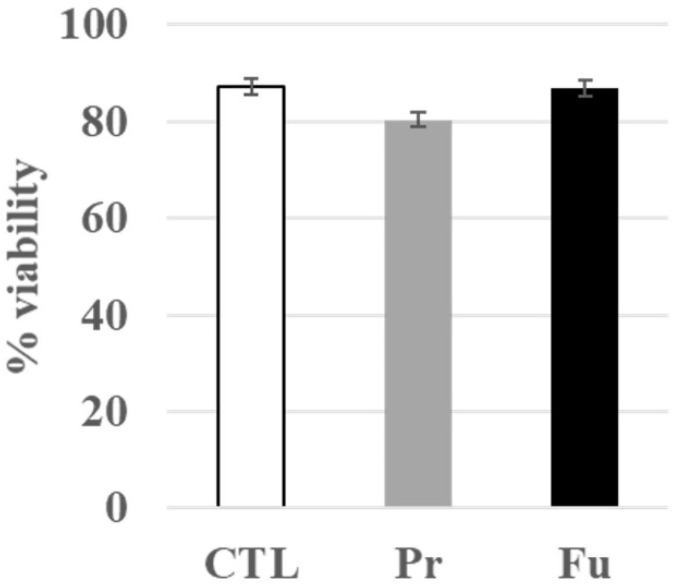
Viability of J774A.1 cells after a 10-day exposure to silica (2 µg/mL/day). CTL: control, cells without nanomaterials, Pr: precipitated silica 220 m^2^/g, Fu: fumed silica 200 m^2^/g.

**Figure 2 nanomaterials-12-01424-f002:**
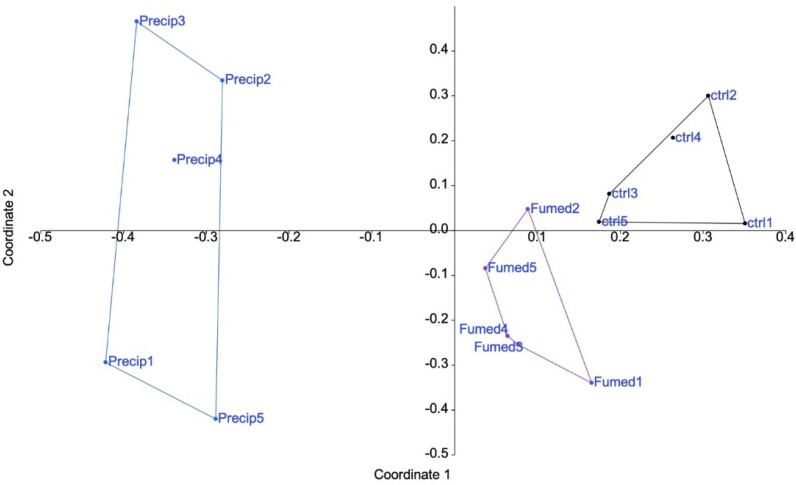
Proteomic results of J774A.1 cells after a 10-days exposure to silica. CTL: control, cells without nanomaterials, Pr: precipitated silica 220 m^2^/g, Fu: fumed silica 200 m^2^/g.

**Figure 3 nanomaterials-12-01424-f003:**
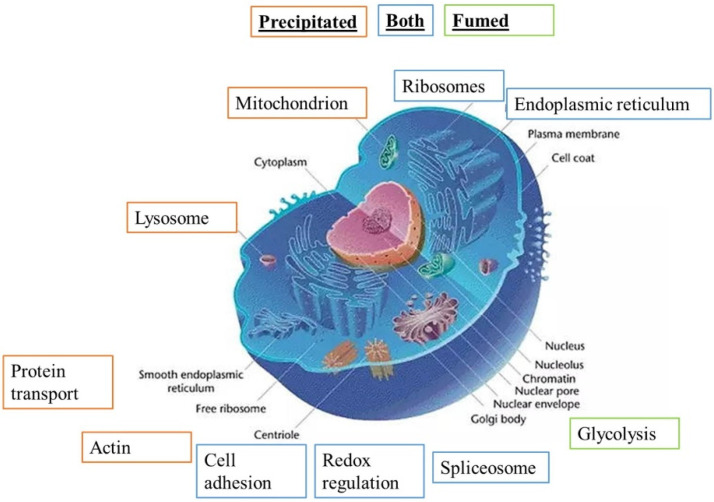
Modified pathways observed with proteomic study (DAVID analysis), adapted from Common Wikimedia Orange: pathway highlighted for precipitated SAS, green: pathway highlighted for fumed SAS, and blue: pathway highlighted for both SAS.

**Figure 4 nanomaterials-12-01424-f004:**
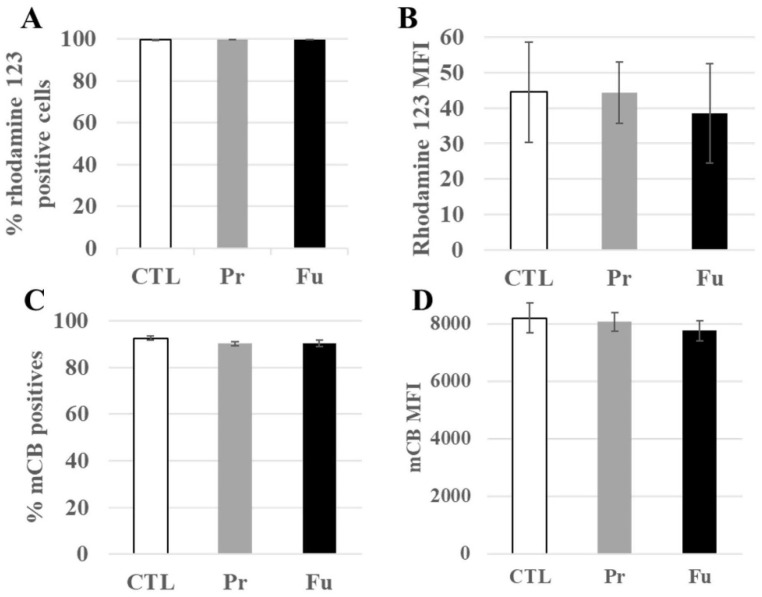
Mitochondrial transmembrane potential and redox potential of J774A.1 cells after a 10-day exposure to silica. (**A**) Proportion of rhodamine 123 positive cells, (**B**) MFI of rhodamine 123 positive cells, (**C**) Proportion of positive cells to monochlorobimane (mCB), (**D**) MFI of mCB+ cells. CTL: control, cells without nanomaterials, Pr: precipitated silica 220 m^2^/g, Fu: fumed silica 200 m^2^/g.

**Figure 5 nanomaterials-12-01424-f005:**
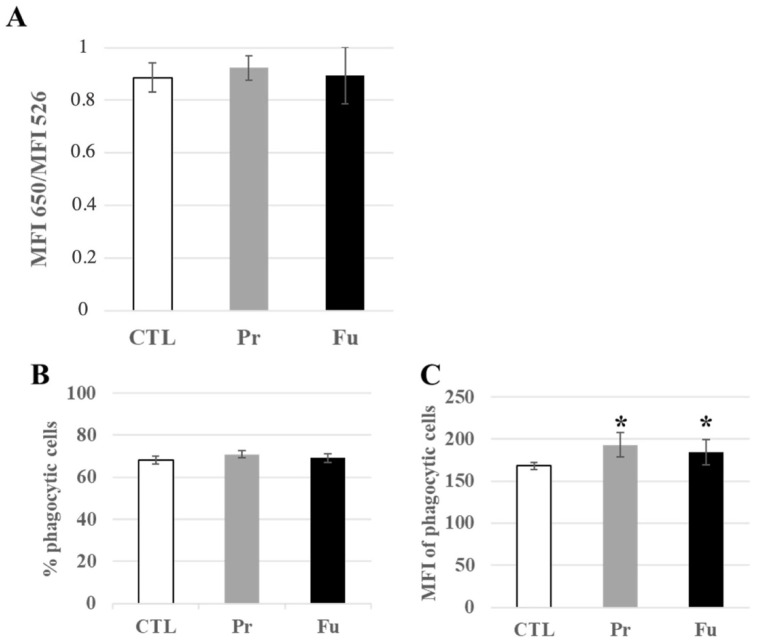
Lysosomal and phagocytic activity of J774A.1 cells after a 10-days exposure to silica. (**A**) Lysosomal ratio using acridine orange (entire/altered lysosomes), (**B**) Proportion of phagocytic cells, (**C**) Activity of phagocytic cells, MFI: mean fluorescent intensity. CTL: control, cells without nanomaterials, Pr: precipitated silica 220 m^2^/g, Fu: fumed silica 200 m^2^/g. Statistical significance in a Student *t*-test: * *p* < 0.05.

**Figure 6 nanomaterials-12-01424-f006:**
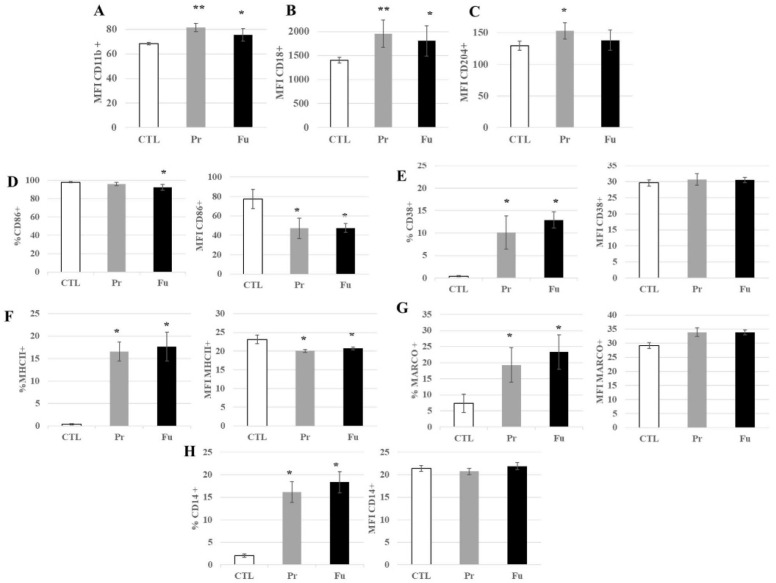
Expression of surface markers of J774A.1 cells after a 10-days exposure to silica. (**A**) CD11b marker (monocyte-macrophage, integrin α), (**B**) CD18 marker (integrin β), (**C**) CD204 marker (scavenger receptor), (**D**) CD86 marker (B7 family), (**E**) CD38 marker (inflammation, induced by IFNγ and LPS), (**F**) MHC class II (inflammation, signalization with CD4 T lymphocytes), (**G**) MARCO marker (scavenger receptor), (**H**) CD14 marker (LPS receptor), (**A**–**C**) All the conditions expressed these markers; only the amount of marker detected at the cell surface is different for cells exposed to SiO_2_. CTL: control, cells without nanomaterials, Pr: precipitated silica 220 m^2^/g, Fu: fumed silica 200 m^2^/g. Statistical significance in a Student *t*-test: * *p* < 0.05; ** *p* < 0.01.

**Figure 7 nanomaterials-12-01424-f007:**
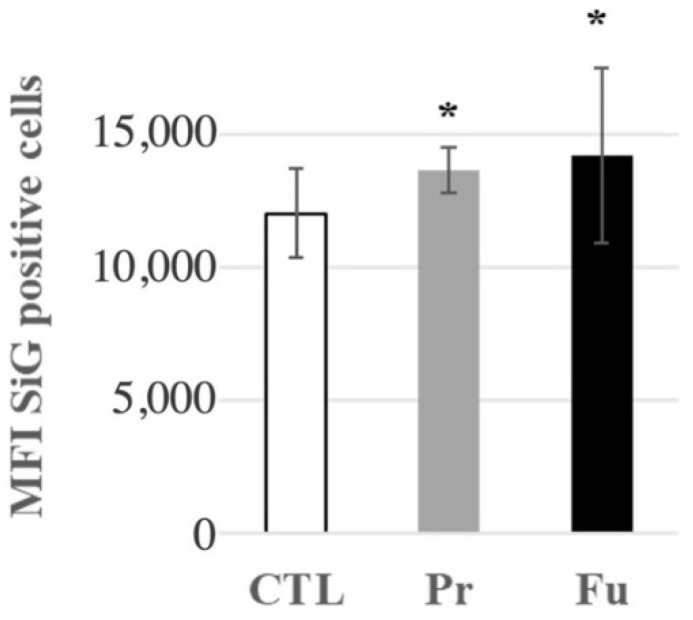
Fluorescent green silica (SiG) internalization of J774A.1 cells after a 10-day exposure to silica. Cells were exposed for 24 h to SiG (days 9 to 10). CTL: control, cells without nanomaterials, Pr: precipitated silica 220 m^2^/g, Fu: fumed silica 200 m^2^/g. Statistical significance in a Student *t*-test: * *p* < 0.05.

**Figure 8 nanomaterials-12-01424-f008:**
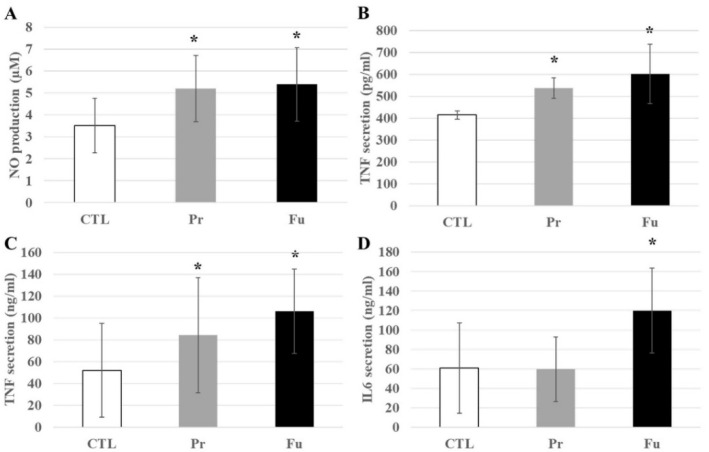
Inflammatory responses of J774A.1 cells after a 10-day exposure to silica, stimulated or not with LPS. (**A**) NO production without LPS, (**B**) TNFα secretion without LPS, (**C**) TNFα secretion with LPS, (**D**) IL-6 secretion with LPS. CTL: control, cells without nanomaterials, Pr: precipitated silica 220 m^2^/g, Fu: fumed silica 200 m^2^/g. Statistical significance in a Student *t*-test: * *p* < 0.05.

**Table 1 nanomaterials-12-01424-t001:** SiO_2_ internalization of J774A.1 cells after a 10-days exposure to silica by ICP-AES dosage, compared with acute exposure.

Si Amount µg/Well (Acute Exposure)	Si Amount µg/Well (Repeated Exposure)	Condition
1.06 ± 0.2	1.6 ± 0.2	Control
36.3 ± 4.4	10.9 ± 4.1	Precipitated silica
27.2 ± 1.1	5.8 ± 0.4	Fumed silica

## Data Availability

Mass spectrometry data are available via ProteomeXchange with the identifier PXD03002 with the doi:10.6019/PXD030002.

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
