# Peer review of "Repeated Exposure of Macrophages to Synthetic Amorphous Silica Induces Adaptive Proteome Changes and a Moderate Cell Activation"

_nanomaterials, 2022, doi:10.3390/nano12091424_

Round 1

Reviewer 1 Report

Overall, I think this is a valuable contribution regarding toxicological findings of SAS materials, especially the proteomics approach can help to uncover new damage mechanisms or cell maintenance mechanisms. Quite important and performed in this work, is the quantification of silica during in vitro SAS testing. In the case of silica, quantification is not trivial and therefore could not be performed and presented very often so far. In the heading of Table 1, an internalization of SiO2 into the cells is mentioned, but this is not evident in the data to be found in this work. In my opinion, more differentiated statements on this or further imaging information, e.g. electron microscopic images, which would prove an uptake of the particles into the cells, are missing.

In my opinion, more detailed characterization data of the two SAS materials are unfortunately missing, especially with regard to their use with FBS. Protein corona formation will affect particle parameters, such as particle dispersion stability or gravitational settling behavior. This in turn has influence on the cellular uptake of the materials and receives too little attention in this work. Further information regarding particle characterizations can be found only insufficiently in the given work (e.g. no information about precipitatd SAS from Solvay, numerical error at item number for fumed silica). Further information on the preparation of the particle dispersions, e.g. dilution factor, would also be helpful, as it is not certain whether it would be necessary to carry a vehicle control. It is pointed out that for good reasons forced diaggregation is not performed, so in my opinion the description of the aggregation state in the given conditions is all the more important. In the discussion it is correctly mentioned that also the quantitative uptake and thus the cellular response of the two forms of SAS could result from different settling behavior. The theory of dissolution or vomocytosis of silica in repeatedly-exposed cells is a good idea that could be tested, e.g., by quantitative determination of silica in the supernatants of the test vials.

In addition, there is no information about proliferation of macrophages during the 10-day test period. According to information in the Cell Culture section, cells are incubated in particle-containing medium with 10%FBS during the tests. Was proliferation not prevented, at least by starvation of the cells with reduced serum content? It is surprising that the tests were performed in this way, since the authors published a long-term test system at the end of last year, in which 1% Horse serum was used to reduce proliferation (even there, proliferation is reduced i.G. to 10% FBS but in my opinion still too clearly present, in `A Low-Serum Culture System for Prolonged in Vitro Toxicology Experiments on a Macrophage System`). I think it would be necessary to verify these experimental conditions, microscopic image data for example could be helpful there. If the cells in the wells grow e.g. too densely or already in several layers (because they are already confluent at the beginning), viability results and also the results for the quantified uptake of the SAS could be falsified.

Overall, I think the reader will understand the approach why in vitro cells should be repeat-exposed. However, the 10-day approach seems to me rather contrived and, due to the lack of effects, also questionable with regard to the detection of new damage mechanisms by SAS. However, the result of differential adaptive proteomic changes involved in the maintenance of cellular homeostasis and the evidence for macrophage polarization are interesting and novel.

Author Response

Dear Reviewer

thank you very much for hour work on our manuscript. Please find below your comments with our replies

Comments and Suggestions for Authors
Overall, I think this is a valuable contribution regarding toxicological findings of SAS materials, especially the proteomics approach can help to uncover new damage mechanisms or cell maintenance mechanisms. Quite important and performed in this work, is the quantification of silica during in vitro SAS testing. In the case of silica, quantification is not trivial and therefore could not be performed and presented very often so far. In the heading of Table 1, an internalization of SiO2 into the cells is mentioned, but this is not evident in the data to be found in this work. In my opinion, more differentiated statements on this or further imaging information, e.g. electron microscopic images, which would prove an uptake of the particles into the cells, are missing.
Comment 
In my opinion, more detailed characterization data of the two SAS materials are unfortunately missing, especially with regard to their use with FBS. Protein corona formation will affect particle parameters, such as particle dispersion stability or gravitational settling behavior. This in turn has influence on the cellular uptake of the materials and receives too little attention in this work. Further information regarding particle characterizations can be found only insufficiently in the given work (e.g. no information about precipitated SAS from Solvay, numerical error at item number for fumed silica). Further information on the preparation of the particle dispersions, e.g. dilution factor, would also be helpful, as it is not certain whether it would be necessary to carry a vehicle control. It is pointed out that for good reasons forced diaggregation is not performed, so in my opinion the description of the aggregation state in the given conditions is all the more important. In the discussion it is correctly mentioned that also the quantitative uptake and thus the cellular response of the two forms of SAS could result from different settling behavior. The theory of dissolution or vomocytosis of silica in repeatedly-exposed cells is a good idea that could be tested, e.g., by quantitative determination of silica in the supernatants of the test vials.
Reply
Regarding the characterization, TEM images are now presented in Figure S1, including images obtained after contact with serum, as is the case in culture medium (and biological media).
Regarding the vomocytosis hypothesis, we have performed an ICP assay on cell supernatants after exposure to silica, and we could not detect silica in these samples (Figure S5). This cannot rule out the hypothesis, as this may be simply due to a lack of sensitivity (see also lines 610-613)
Comment
In addition, there is no information about proliferation of macrophages during the 10-day test period. According to information in the Cell Culture section, cells are incubated in particle-containing medium with 10%FBS during the tests. Was proliferation not prevented, at least by starvation of the cells with reduced serum content? It is surprising that the tests were performed in this way, since the authors published a long-term test system at the end of last year, in which 1% Horse serum was used to reduce proliferation (even there, proliferation is reduced i.G. to 10% FBS but in my opinion still too clearly present, in `A Low-Serum Culture System for Prolonged in Vitro Toxicology Experiments on a Macrophage System`). I think it would be necessary to verify these experimental conditions, microscopic image data for example could be helpful there. If the cells in the wells grow e.g. too densely or already in several layers (because they are already confluent at the beginning), viability results and also the results for the quantified uptake of the SAS could be falsified.
Reply
The cells are used at confluence, and macrophages cell lines do not grow in several layers, while other cell types (e.g. hepatocytes) do. Thus the only problem that we faced with the culture system in high serum is cell detachment, which can be massive and which would completely bias the results, as mentioned by the reviewer. We now detail in lines 116-120 how we took this problem into account, and discuss it in lines 600-603.

Comment
Overall, I think the reader will understand the approach why in vitro cells should be repeat-exposed. However, the 10-day approach seems to me rather contrived and, due to the lack of effects, also questionable with regard to the detection of new damage mechanisms by SAS. However, the result of differential adaptive proteomic changes involved in the maintenance of cellular homeostasis and the evidence for macrophage polarization are interesting and novel.
Reply
We are glad that the reviewer shares the interest for repeated exposures. The difficulty that arises in repeated exposures in vitro is the balance between dose division, experimental speed and maintenance of the performances of the in vitro system.  Consequently, the 10 days exposures is of course a compromise. If we are intellectually honest, the same criticism applies to in vivo system, in the sense that we postulate that a 28 days experiment in rodents is equivalent to years in humans. This is already an approximation, although of lesser magnitude than for in vitro systems. 
Contenting the damage mechanisms by SAS, we would have loved to have new mechanisms to report, as every toxicologist. However, we also feel that « negative » results must be reported. 

Reviewer 2 Report

The paper, entitled "Repeated exposure of macrophages to synthetic amorphous silica induces adaptive proteome changes and moderate cell activation," aims to use instead a repeated 10-day exposure regimen in an effort to better simulate the occupational exposures to which consumers and workers are exposed in daily life. The authors used the murine macrophage cell line J774A.1 as a biological model because macrophages are very important innate immune cells in the response to particulate materials. To better assess macrophage responses to repeated exposure to SAS, the authors used proteomics as a broad-based approach. The authors screened out several candidate proteins to test the quantification of these candidates. They found that intracellular glutathione levels or mitochondrial transmembrane potential at the end of the 10-day exposure were similar in SAS-exposed cells and unexposed cells. These results suggest that repeated exposure to low doses of SAS slightly modulates the immune functions of macrophages, which could alter the homeostasis of the immune system. The experimental design and data analysis of this project are comprehensive. This work provides new data. It is worthy of publication.

Author Response

Dear Reviewer

Thank you very much for your positive appreciation of our work

Reviewer 3 Report

The manuscript aims to study the effects of synthetic amorphous silica (precipitated and fumed silica) on murine macrophages upon repeated exposure. The authors performed a proteomic analysis then validated through different and targeted methodological approaches. The manuscript is quite well written, presented and discussed. I do not observe relevant limitations but I have some important comments that I try to summarise: 

  1. the authors highlight the concept that they use a repeated exposure scheme to better simulate the occupational exposure encountered in daily life by consumers and workers, and they declare that they use a realistic dose of silica. Is it realistic for humans? Why did they use murine cell model?  The authors should comment the limit of the study in this sense considering that they are neither using in vivo model of exposure nor human primary cells. In any case a validation on human primary cells could highly increase the value of the manuscript.
  2. The figure 2 shows the PCA analysis and it is evident how different samples (control, fumed and precipitated) clustered separately. However, there is an important variability between replicates (I understood that the numbers indicate the replicates). This variability is evident also in terms of standard deviations observed in the other analysis and figures, such as figures 5, 6, and 7. How do the authors explain this weak repeatability since they used a model based on cell line?
  3. Please, add a figure for DAVID analysis in which more significant biological pathways are shown. 
  4. Please, check the sentence lines 329-332. 

Author Response

Dear Reviewer

thank you for your work on ur manuscript. Please find below your comments and our replies

Comment
1. the authors highlight the concept that they use a repeated exposure scheme to better simulate the occupational exposure encountered in daily life by consumers and workers, and they declare that they use a realistic dose of silica. Is it realistic for humans? Why did they use murine cell model?  The authors should comment the limit of the study in this sense considering that they are neither using in vivo model of exposure nor human primary cells. In any case a validation on human primary cells could highly increase the value of the manuscript.

Reply
The issues risen by the reviewer in this comment are very relevant. Some key points for using in vitro methods are the ethical issue of using less laboratory animals, the speed of the methods and their reduced costs. This means that when trying to mimic occupational exposure, the in vitro systems must be able to withstand the time contraints, i.e. the ability to survive long enough to allow the repeated exposure to be carried out. Most unfortunately, this is not the case of primary macrophages which cannot be kept alive in vitro for such extended periods of time. Indeed the system that we used is, for the time being, the only one that could be used for repeated exposures.

Comment
2.The figure 2 shows the PCA analysis and it is evident how different samples (control, fumed and precipitated) clustered separately. However, there is an important variability between replicates (I understood that the numbers indicate the replicates). This variability is evident also in terms of standard deviations observed in the other analysis and figures, such as figures 5, 6, and 7. How do the authors explain this weak repeatability since they used a model based on cell line?

Reply
The variability observed by the reviewer is real and is linked to two phenomena. One is specific to proteomics and especially to shotgun proteomics. In shotgun proteomics the emphasis is put on measuring simultaneously the highest numbers of proteins, i.e. the highest numbers of peptides. This comes at the price of the quality of the measurements, so that the measurements made for each and every protein are not very precise. Furthermore, the reconstruction of a protein abundance from abundances of peptides is not a straightforward process, adding further variability. 
In addition to that, 10 days in culture represent a long time, even for a robust in vitro system. There is thus some variability that arises because of the long culture time, compared to classical 24 hours exposures.

    3.    Please, add a figure for DAVID analysis in which more significant biological pathways are shown. 

This is now done in Figure 3, and cited in line 388

Comment
4.Please, check the sentence lines 329-332. 

Reply
Thank you for pointing to this rather obscure writing of ours. The two sentences have been modified (now lines 464-465)

Reviewer 4 Report

The authors reported a repeated 10-days and low dose exposure of two synthetic amorphous silica to murine macrophage cell line J774A.1 and used omics method to evaluate the potential activated pathways. The authors then studied the target outcomes derived from the pathway analysis for further discussion, which is quite reasonable and informative. I recommend the publication of the manuscript in Nanomaterials, after the authors considered the following comments and suggestions.

Major:

  1. The authors washed the cell only once with PBS, is the way enough to remove particles which are not attached on the cells or adsorbed on the surface of culture plate? I am doubt that most of Si you’ve detected is not from the cellular uptake. The authors should consider using TEM as supporting tool.
  2. Line 222: The cells were also primed or not with LPS (100 ng/mL) for the last 24 h of culture. The authors should explain why they designed the exposure of LPS like this, and discuss their findings in discussion part.
  3. The authors found that uptake was more pronounced for precipitated silica than for fumed silica in the case of the 10 days exposure. Are the two SASs have different stability in the medium even though both aggregate sizes are in the 200-500 nm?

Minor:

Line 35: MHCII

Line 39: A reference should be added to support the production rate.

Line 50: “This suggests a strong parallel between the toxicology of SAS and the one of crystalline silica.” This sentence confused me, are there many kinds of crystalline silica?

Line 57: Do you mean occasional?  The use of SAS as a food additive or in cosmetics is not occupational exposure.

Line 82, 84 and 97:  2 should be superscript.

Line 115: Which kinds of standards did you buy from sigma should be stated clearly? Line 117: What is qsp?

Line 190: quenching [49], as described previously [50].

Line 238: EC20? LD is generally used for acute in vivo study.

Line 241: Dose information should be added.

Line 529: the functional effects an especially the effects on surface markers and cytokine production were similar for both types of silica. This sentence should be modified.

Line 390: The authors should state why IL-6 data was not shown in no LPS stimulation group in the figure caption.

Author Response

Dear Reviewer

thank you very much for you work on our manuscript. Please find below your comments and our replies

Comments and Suggestions for Authors
The authors reported a repeated 10-days and low dose exposure of two synthetic amorphous silica to murine macrophage cell line J774A.1 and used omics method to evaluate the potential activated pathways. The authors then studied the target outcomes derived from the pathway analysis for further discussion, which is quite reasonable and informative. I recommend the publication of the manuscript in Nanomaterials, after the authors considered the following comments and suggestions.

Major:

Comment
1.The authors washed the cell only once with PBS, is the way enough to remove particles which are not attached on the cells or adsorbed on the surface of culture plate? I am doubt that most of Si you’ve detected is not from the cellular uptake. The authors should consider using TEM as supporting tool.

Reply

TEM experiments  have been performed, backed by confocal microscopy experiments which are not prone to cutting artifacts. The results of these experiments, shown in Figure S2 and S3, indicate that the silica particles were full internalized. This is also mentioned in the main text in lines 351-353

Comment
2. Line 222: The cells were also primed or not with LPS (100 ng/mL) for the last 24 h of culture. The authors should explain why they designed the exposure of LPS like this, and discuss their findings in discussion part.

Reply
A reply to this useful suggestion is now made in lines 121-123 (methods) and 710-716 (discussion)

Comment

3. The authors found that uptake was more pronounced for precipitated silica than for fumed silica in the case of the 10 days exposure. Are the two SASs have different stability in the medium even though both aggregate sizes are in the 200-500 nm?

Reply

This point has been addressed in Figure S1 and is now mentioned in lines 339-341

Minor:
Line 35: MHCII

Done

Line 39: A reference should be added to support the production rate.

Done, ref 1

Comment
Line 50: “This suggests a strong parallel between the toxicology of SAS and the one of crystalline silica.” This sentence confused me, are there many kinds of crystalline silica?

Reply
There are, indeed, but this is not the question. The question is that because of the toxic effects of crystalline silica, the toxicological studies made on amorphous silicas have explored mainly the same phenomena and mechanisms

Comment
Line 57: Do you mean occasional?  The use of SAS as a food additive or in cosmetics is not occupational exposure.

Reply
Thank you for the comment. We have replaced « occupational » by « chronic »

Comment
Line 82, 84 and 97:  2 should be superscript.

Reply
Done

Comment
Line 115: Which kinds of standards did you buy from sigma should be stated clearly? Line 117: What is qsp?
Standards are now more detailed (lines 143 and 146). Qsp is an abbreviation for latine Quantitas quae Sufficit Per , which could be translated by « amount required to reach ». We have changed the wording of the sentence to make it clearer

Comment
Line 190: quenching [49], as described previously [50].

Reply
Corrected

Comment
Line 238: EC20? LD is generally used for acute in vivo study.

Reply
LD is also used for in vitro systems when the parameter tested is viability, which was the case. EC20 is more generic and can apply to any endpoint

Comment
Line 241: Dose information should be added.

Reply
Done, thank you for the comment

Comment
Line 529: the functional effects an especially the effects on surface markers and cytokine production were similar for both types of silica. This sentence should be modified.

Reply
The sentence has been modified (lines 736-740). We hope it will be clearer and more accurate.

Comment
Line 390: The authors should state why IL-6 data was not shown in no LPS stimulation group in the figure caption.

Reply
This is stated in lines 554-555

Round 2

Reviewer 4 Report

The authors address most of my concerns about the article, and can be accepted in "Nanomaterials" after the following format errors.  

1. Please check all of the format in the manuscript to the high scientific standard. There were still many typos. ex. 24 h (somewhere you use 24 hours, some use 24h).  2 µg/ml,  5 pg/ml, 200 m²/g...etc. In figure s1-s3, and please don't use "x", rather use × (symbol) to represent magnification. 

Author Response

Thank you for your appreciation of our manuscript and for your comments. We have combed the manuscript and hope to have corrected all the typos